

# Moisture ELevated Temperature (MELT) index: A novel index to capture dry and humid heatwaves

**Kwesi Twentwewa Quagraine[1], Kwesi Akumenyi Quagraine[2,3]**

[1]Earth and Atmospheric Science Department, Indiana University, Bloomington, Indiana
[2]NSF National Center for Atmospheric Research, Boulder, Colorado
[3]Department of Physics, University of Cape Coast, Cape Coast, Ghana

Corresponding author: K. A. Quagraine, `kwesiq@ucar.edu`





**Abstract**

In this study, we introduce a novel heatwave characterization metric: the Moisture ELevated Temperature (MELT) index. MELT integrates anomalies in temperature and relative humidity to quantify moist heatwaves and more accurately assess physiological heat stress. Traditional heatwave metrics predominantly rely on temperature alone, often underestimating the compounded effects of humidity on human health and thermoregulation. To address this gap, the MELT index offers improved accuracy for public health risk assessment and response strategies. To validate MELT's effectiveness and versatility, we applied it to analyze three significant, record-breaking heatwave events from recent decades: the 2021 Pacific Northwest (PNW), 2016 South Korea, and 2019 Western Europe heatwaves. Our analysis demonstrates that MELT clearly distinguishes between humid and dry heatwave conditions, accurately identifying the moisture characteristics specific to each region. Specifically, the PNW and South Korea events exhibited notably higher humidity levels, influenced by atmospheric rivers and increased convective activities, respectively. Conversely, the Western Europe heatwave was characterized by drier conditions resulting from Saharan dry-air intrusions. MELT's reliance on widely accessible datasets of temperature and humidity ensures its global applicability and consistency, addressing limitations inherent in temperature-only indices. Furthermore, its flexible use of climatological percentile thresholds allows adaptation to varying climates and future scenarios. Given anticipated increases in heatwave frequency and intensity due to climate change, MELT provides a critical tool for evaluating emerging risks, informing climate adaptation policies, and guiding targeted mitigation measures.

## 1 Introduction

Exposure to heat extremes, such as heat stress and heatwaves, represents one of the leading causes of weather-related mortality and morbidity globally, with recent events demonstrating devastating impacts across different regions (Ebi et al., 2021; Sheridan et al., 2021; Weilnhammer et al., 2021; Vicedo-Cabrera et al., 2021; Tuholske et al., 2021; Perkins-Kirkpatrick & Lewis, 2020). While traditional temperature-based metrics have long been used to assess heat risks, mounting evidence suggests that these measures alone may not adequately capture the full physiological burden of heat stress on the human body (e.g., Simpson et al. (2023); Sherwood (2018)). The interaction between temperature and humidity plays a crucial role in determining how heat is actually experienced and its potential health impacts, since elevated moisture levels can significantly affect the body's natural cooling mechanism through sweat evaporation (Sherwood, 2018; Cramer et al., 2022; Ioannou et al., 2022; Luber & McGeehin, 2008).

A moisture-inclusive heat index offers several distinct advantages over purely temperature-based and or empirical-based measures. First, it provides a more accurate assessment of physiological heat stress by accounting for the combined effects of temperature and humidity on human thermoregulation (Spangler et al., 2022; Fiala et al., 2012). Recent research by (Simpson et al., 2023) demonstrates that different heat stress indices (HSIs) can lead to substantially different conclusions about heat risk, particularly in conditions where humidity plays a significant role. Second, such indices better inform public health responses by enabling more precise risk assessment and targeted interventions for vulnerable populations (Baldwin et al., 2023).

The limitations of traditional temperature-only approaches have become increasingly apparent as climate change drives both rising temperatures and changes in humidity patterns. As highlighted by (Baldwin et al., 2023) and (Steadman, 1979), while epidemiological studies have historically focused primarily on temperature, the compound effects of heat and humidity create distinct challenges for human health that cannot be captured by temperature alone. This is particularly relevant for urban areas and regions experiencing both high



temperatures and humidity levels, where traditional metrics can underestimate actual heat stress (Buzan & Huber, 2020; Cvijanovic et al., 2023).

From an operational perspective, moisture-inclusive indices provide valuable tools for workplace safety, public health planning, and emergency response. Studies have shown that metrics that incorporate humidity, such as wet-bulb globe temperature (WBGT) and universal thermal climate index (UTCI), can predict more effectively heat-related health outcomes compared to temperature alone (Ioannou et al., 2022; Cvijanovic et al., 2023; Fiala et al., 2012). These comprehensive measures are especially crucial for outdoor workers, athletes, and vulnerable populations who may be exposed to varying combinations of heat and humidity (Vecellio et al., 2022; Tuholske et al., 2021; Rey et al., 2009).

However, the adoption of moisture-inclusive indices faces several challenges, including the need for standardization across different regions and applications, as well as the complexity of communicating these measures to the public (Davis et al., 2016). Recent research emphasizes the importance of selecting appropriate indices based on specific contexts and applications while also considering the need for clear communication of heat risks to diverse audiences (Simpson et al., 2023; Perkins-Kirkpatrick & Lewis, 2020). Despite these challenges, the growing evidence supporting the advantages of moisture-inclusive heat indices suggests their increasing importance in addressing heat-related health risks in a warming world (Simpson et al., 2023; Cvijanovic et al., 2023). In this paper, we propose a standardized index that is applicable in various regions and climates. This index utilizes temperature and relative humidity, readily available atmospheric variables, to compute a Moisture ELevated Temperature (MELT) index. Previous research on heat stress has often focused on developing indices tailored to specific regions or relying on empirical values that are region-specific and not easily scalable across different time scales. Our method addresses this limitation by offering a more universally applicable approach.

## 2 Methods

The *Moisture Elevated Temperature (MELT) Index* quantifies moist heatwaves by integrating temperature and relative humidity anomalies (see Eq. 1). It consists of two components. First, a grid point is classified as experiencing a *heatwave* if the daily maximum 2m temperature exceeds its 95[th] percentile (computed from the 1990–2021 reference period) for at least three consecutive days. This classification is represented by a binary indicator: 1 for a heatwave and 0 otherwise.

The second component evaluates how relative humidity compares to its respective 95[th] percentile over the same period. This ratio provides insight into how anomalously high humidity levels contribute to extreme heat conditions. The MELT index is then defined as the product of the heatwave indicator and the relative humidity ratio, effectively capturing moisture-amplified heat extremes (see Eq. 2). This novel metric enhances the characterization of humid heatwaves that exceed the 95[th] percentile, offering a refined approach for studying extreme heat events. To evaluate the ability of this index, we further apply the proposed index to different extreme events in the present climate (1990-2021), to assess its ability to capture these events comparing to literature as a means of evaluation.

**Moisture Elevated Temperature (MELT) Index**:

$$H(t) = W(t) \cdot \frac{RH(t)}{RH_{95}} \tag{1}$$

The heatwave condition (W(t)), is defined mathematically below:

$$W(t) = \begin{cases} 1, & \text{if } T'(t) > 1 \text{ for } t, t+1, t+2 \\ 0, & \text{otherwise.} \end{cases} \tag{2}$$





where:

- $H(t)$: Heat index on day $t$
- $T(t)$: Observed temperature on day $t$
- $RH(t)$: Observed relative humidity on day $t$
- $T_{clim}$: 95$^{th}$ percentile 2m temperature
- $RH_{clim}$: 95$^{th}$ percentile RH
- $T'(t)$ : $\frac{T(t)}{T_{95}}$

Thus, $W(t) = 1$ (indicating a heatwave) if $H(t) > 1$ for three or more consecutive days, and $W(t) = 0$ otherwise. The unitless index $H(t)$ represents the magnitude of the heatwave and its associated relative humidity (RH). This formulation allows for assessing the contribution of RH to heatwave events. Higher MELT values indicate increased RH, signifying moist heatwaves, while lower MELT values correspond to drier heatwaves. A MELT value of 0 indicates the absence of a heatwave, and thus RH its not considered.

### 2.1 Synoptic, dynamics and thermodynamics drivers associated with MELTS

To evaluate the MELT index's ability to capture moisture-enhanced heatwaves, we computed MELT values for three distinct heatwave events occurring in different regions globally, aiming to address the question: "What role does moisture play in these heatwaves?". Specifically, if a heatwave event coincides with surface moisture (relative humidity) at or above its 95$^{th}$ percentile, we classify it as a moisture-enhanced (humid) heatwave.

Next, we analyzed the synoptic-scale dynamics associated with these MELT events using data from the ECMWF Reanalysis 5 (ERA5) dataset (Hersbach et al., 2020). Variables used in this analysis include maximum surface temperature, specific humidity, geopotential heights, and wind vectors. Our analysis further investigated the synoptic conditions underpinning these events by posing the question: "Do moisture-enhanced heatwaves have stronger heat domes?". A heat dome is typically defined as a region characterized by anomalous anticyclonic circulation in the mid-to-upper troposphere coupled with anomalously high surface temperatures (National Environmental Satellite, Data, and Information Service, 2023).

For the present-day evaluation of heatwave events, we computed climatological thresholds based on a reference period from 1990 to 2021. Specifically, the 95$^{th}$ percentile values for temperature and relative humidity were calculated over this baseline period. This climatological interval can be adjusted to suit analyses of different climatic contexts or time frames.

## 3 Results and Discussions

We assess three heatwave events using the MELT index to determine its ability to capture these events and illustrate the associated synoptic-scale dynamics before, during, and after each event. Below are some key considerations for interpreting results obtained with the MELT index:

- If an event has a MELT index $> 0$, then, there is a heatwave.
- If an event has a MELT index $> 0$ and $< 1$, then, there is a heatwave with humidity below the 95$^{th}$ percentile of the climatological relative humidity.
- If an event has a MELT index $< 0.5$ then, there is a relatively dryer heatwave with humidity lower than 50% of the climatological relative humidity.
- If an event has a MELT index $>= 1$, then, there is a heatwave with humidity greater than or equal to the 95$^{th}$ percentile of climatological relative humidity.





### 3.1 Pacific Northwest (PNW) Heatwave Event

The PNW heatwave occurred in late June 2021 (Neal et al., 2022; Wang et al., 2023) across the Pacific Northwest of North America, with temperatures soaring as high as 45°C in a region typically characterized by a cool climate (Neal et al., 2022; Schumacher et al., 2022). During this event, observed maximum temperatures exceeded even the upper bound estimates, including those adjusted for anthropogenic climate change, indicating that such an extreme could not have been predicted under standard univariate extreme value analysis assumptions (Zhang et al., 2024).

This record-breaking heatwave was driven by a strong atmospheric blocking pattern and upstream cyclogenesis, which contributed to a stratified atmosphere conducive to heat trapping (Schumacher et al., 2022; Neal et al., 2022). Additionally, a record-breaking mid-tropospheric ridge, anomalously high mid-tropospheric temperatures, strong subsidence, and drier-than-normal soil moisture played crucial roles in intensifying the event (Mass et al., 2024). Literature further suggests that the anticyclonic circulation aloft converted previously stored potential energy into sensible heat through subsidence, leading to escalating near-surface temperatures (Schumacher et al., 2022).

In this case study, we evaluate the ability of MELT to capture this unprecedented heatwave and analyze its surrounding synoptic-scale meteorological conditions.

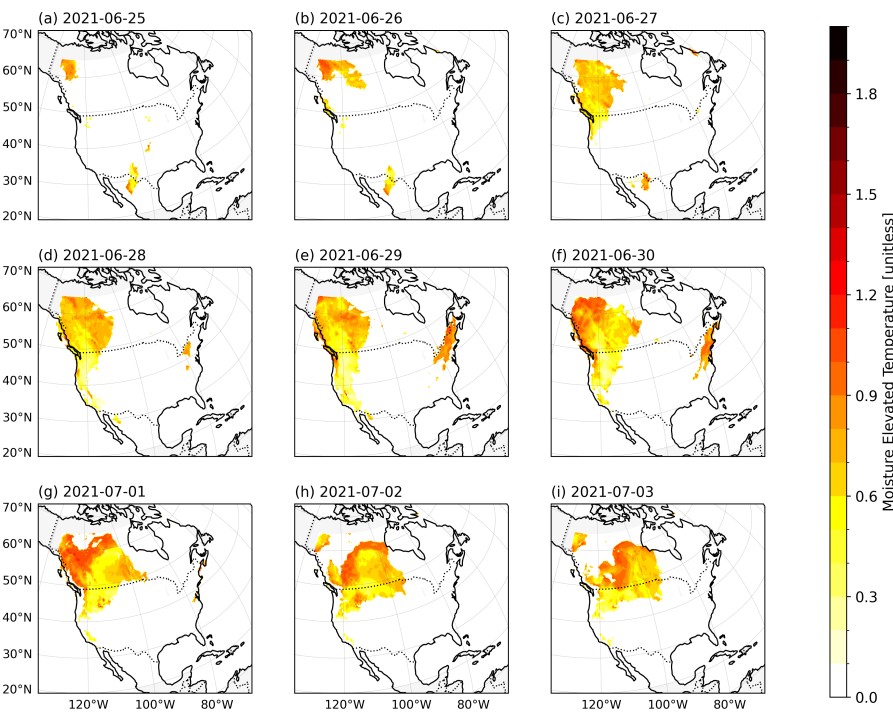

**Figure 1.** Moisture ELevated Temperature (MELT) Index for the Pacific Northwest heatwave in late June 2021 to early July, 2021. Figure shows the MELT index for the heatwave.





Our results indicate that the MELT index effectively captures the PNW heatwave event. Additionally, our findings align with the regional extent of the heatwave reported in recent literature (Zhang et al., 2024; Mass et al., 2024). We observe that moisture levels are significantly elevated (MELT $\geq$ 1) primarily from June $30^{th}$ through July $1^{st}$, 2021. Subsequently, these high-moisture conditions gradually shift away from the PNW region, moving eastward into the northern continental United States (CONUS) and southern Canada.

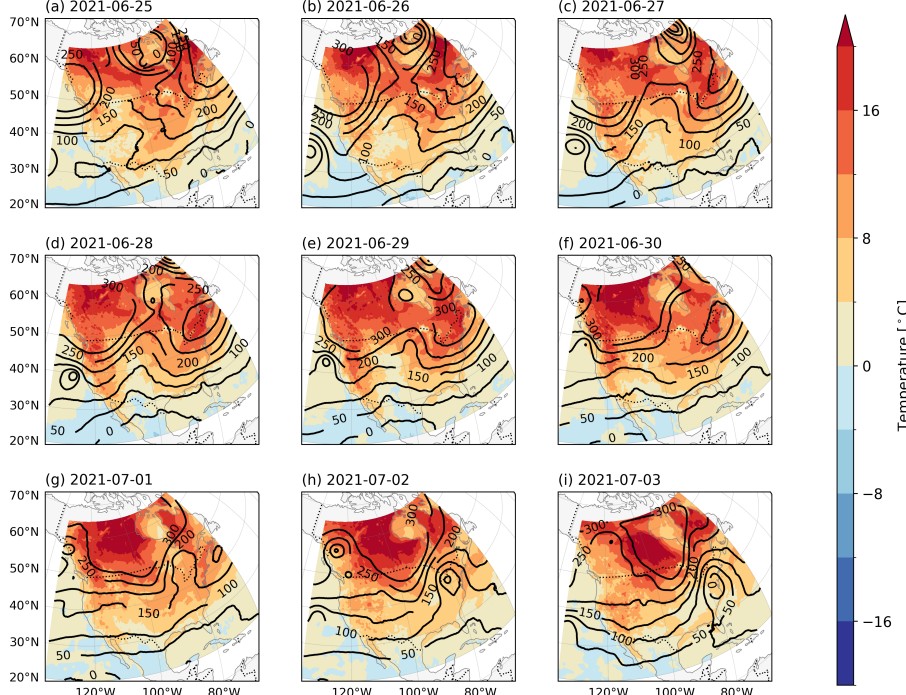

**Figure 2.** **Synoptic scale anomalous maximum surface temperature and geopotential height contours (in meters) at 500 hPa composites for the Pacific Northwest heatwave in late June 2021 to early July, 2021**

The synoptic-scale weather conditions during the event (Fig. 2) reveal a prominent anomalous anticyclonic circulation entering the region, characterized by geopotential height anomalies as high as 100 m, indicative of mid-tropospheric descent. At the surface, anomalously high temperatures ( 20°C) were observed. Our composite analysis aligns closely with patterns reported in previous studies, including those by Schumacher et al. (2022); Bercos-Hickey et al. (2022). Additionally, we identify anomalously high moisture deposition ( 10 kg/kg) within the region during the event (Fig. 3). This finding is consistent with the analysis by Mo et al. (2022), who attribute the high humidity to an active atmospheric river (AR) event. Their study highlights that this AR transported significant moisture and heat from Southeast Asia into the northeastern Pacific region, further intensifying humidity levels.

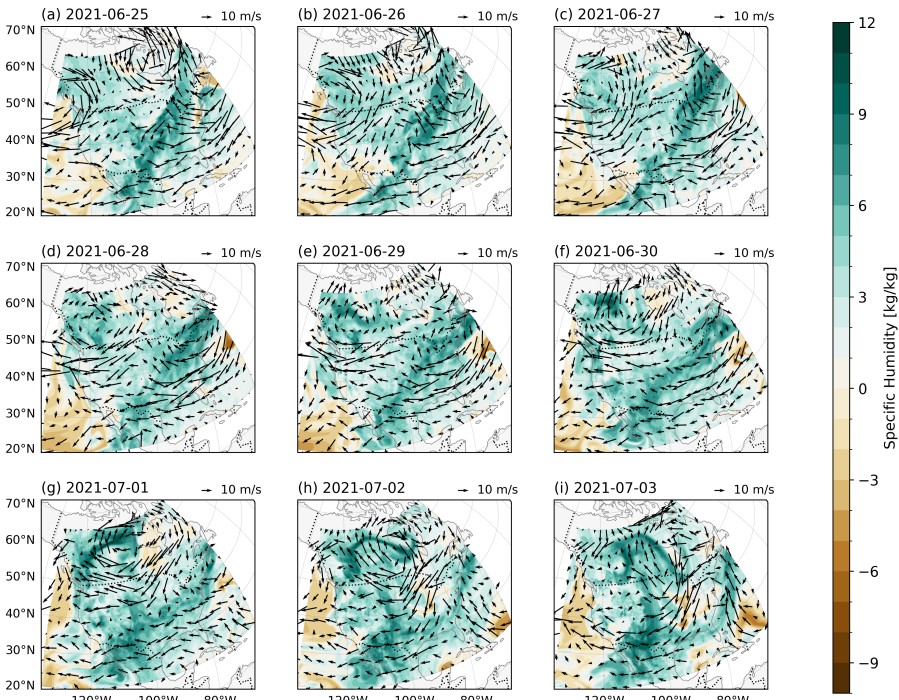

**Figure 3. Synoptic scale composites for anomalous near-surface (850 hPa) specific humidity and wind circulation (500 hPa $\vec{U}$) for the Pacific Northwest heatwave in late June 2021 to early July, 2021**

### 3.2 West Europe Heatwave

This heatwave occurred during the last week of June 2021 in western Europe (Froment & Below, 2020), breaking several historical records with temperatures reaching as high as 46°C near the city of Nîmes. According to Vautard et al. (2020), human-induced warming played a significant role in this event, noting that such short but intense heatwaves would have been extremely unlikely (a 1-in-1000-year occurrence) without anthropogenic climate change. Additionally, they highlight that similar events in a pre-industrial climate would have been approximately 1.5°C to 3°C cooler. Furthermore, Sousa et al. (2020) demonstrate that strong dry-air intrusions from the northern Sahara, combined with low regional soil moisture, significantly intensified the heatwave, leading to unprecedented temperatures. In contrast to the Pacific Northwest (PNW) heatwave, this event was notably drier.

Our results demonstrate that the MELT index (Fig. 4) effectively captures this heatwave event and provides insights into its humidity characteristics. Consistent with findings by Sousa et al. (2020), we observe relatively drier conditions near southwestern Europe, resulting from dry air intrusion originating from the northern Sahara. Conversely, regions closer to coastal water bodies along the northern flank exhibit higher moisture content compared to inland areas.



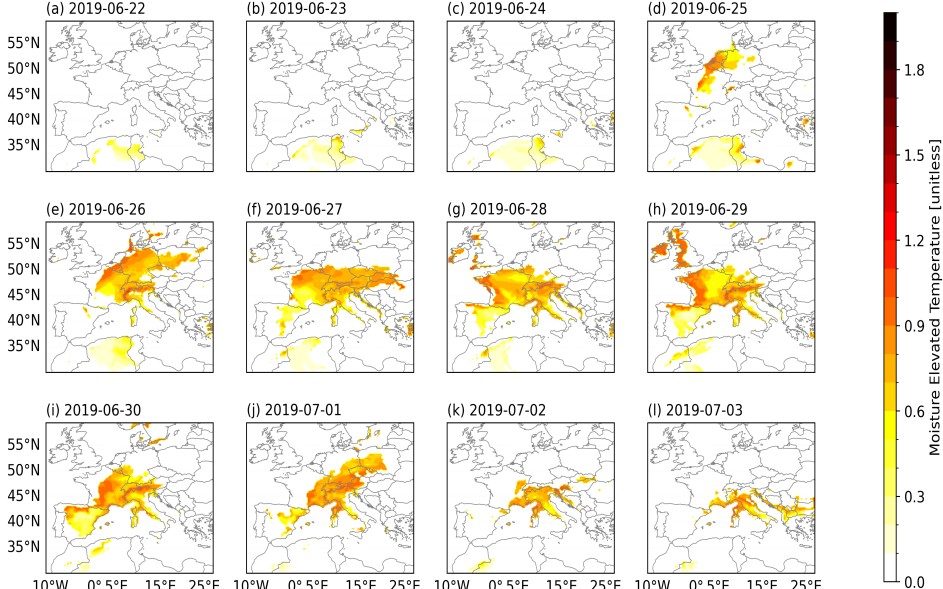

**Figure 4.** **Moisture ELevated Temperature (MELT) Index for the west Europe heatwave in late June 2019 to early July, 2019. Figure shows the MELT index for the heatwave.**

The analysis also identifies the peak MELT index occurring on June $29^{th}$, 2019. During this peak period (June $28^{th}$–$29^{th}$, 2019), heatwaves over regions such as Spain and northeastern France remain relatively dry, while coastal France experiences more humid conditions. Similarly, coastal areas of the United Kingdom (UK) and Ireland show a progressive increase in humidity throughout the duration of the heatwave event.

During the West European heatwave, unprecedented high temperatures occurred across the region beginning on June $25^{th}$. However, this initial period does not meet the three-consecutive-day criterion necessary to classify it as a heatwave. During this event, an anomalous high-pressure system became stationary in the mid-troposphere, trapping warm air along with dry-air intrusion brought by southerly winds.

Consequently, anomalously dry conditions dominated the region, although pockets of humid air persisted near coastal areas. Over areas experiencing the most intense heatwave conditions, wind vectors reveal the development and progression of anticyclonic circulation, intensifying and becoming well-defined by June $29^{th}$ before dissipating afterward. Our composite analysis, utilizing the MELT index, effectively captures this heatwave, highlighting its thermodynamic properties and dynamic synoptic-scale meteorological features.

### 3.3 South Korea Heatwave

The heatwave that struck South Korea in August 2016 persisted nearly the entire month. Throughout this period, an anomalously high geopotential height was consistently observed over Mongolia, coinciding with exceptionally high surface temperatures. This anomalous high-pressure system facilitated the inflow of warm northerly winds onto the




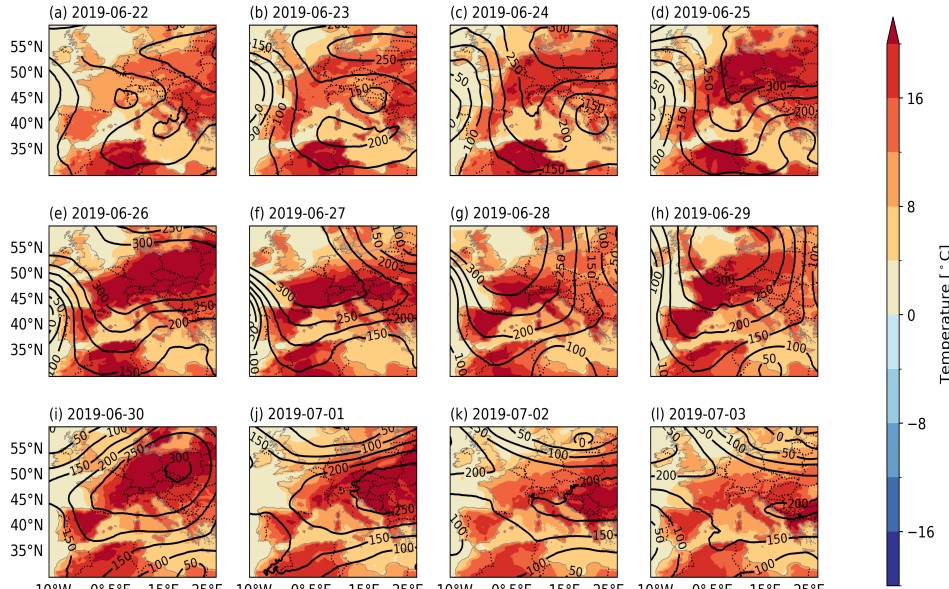

**Figure 5.** **Synoptic scale anomalous maximum surface temperature and geopotential height contours (in meters) at 500 hPa composites for the west Europe heatwave in late June 2019 to early July, 2019.**

Korean Peninsula (Yeh et al., 2018). Additionally, Yeh et al. (2018) attributed the unusual severity of this heatwave to strong convection over the western-to-central subtropical Pacific, transmitted through atmospheric teleconnections. In this study, we apply the MELT index to evaluate the moisture content and duration of the event, while also analyzing the associated synoptic-scale meteorological conditions.

The MELT index (Fig. 7) indicates moisture played a pivotal role in this heatwave. MELT index values exceeding 1 are observed across most locations, aligning with the findings of (Yeh et al., 2018), who demonstrated that convective activity was a key driver of this event. The heatwave persists primarily over mainland South Korea and extends to coastal areas near the Philippine Sea.

The maximum surface temperature (Fig. 8) shows that while temperatures over land are high, ocean temperatures are comparatively higher. The presence of an anomalous high-pressure system in the mid-troposphere (approximately 300 hPa), overlaying regions with anomalously high maximum surface temperatures (around 15°C), creates conditions resembling a heat dome. This configuration contributes significantly to elevated MELT index values across the region. Additionally, moisture content is anomalously high (¿10 kg/kg) during this heatwave, coinciding with a spatially extensive anomalous anticyclone. This combination results in notably humid conditions during the heatwave period (Fig. 9).

## 4 Conclusions

In this study, we propose a novel heatwave characterization metric: the Moisture ELevated Temperature (MELT) index. MELT quantifies moist heatwaves by integrating



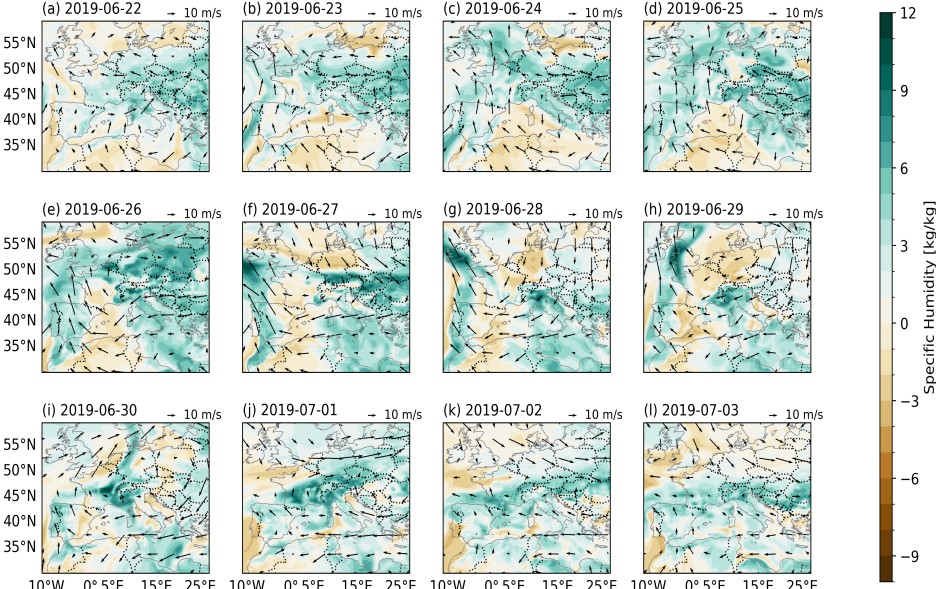

**Figure 6.** **Synoptic scale composites for anomalous near-surface (850 hPa) specific humidity and wind circulation (500 hPa $\vec{U}$) for the west Europe heatwave in late June 2019 to early July, 2019.**

anomalies in temperature and relative humidity, providing a more accurate assessment of physiological heat stress. This approach captures the combined effects of temperature and humidity on human thermoregulation and thus enables better quantification of associated health risks, helping to inform public health responses. To evaluate the robustness and versatility of the MELT index, we applied it to several notable, record-breaking heatwave events from the past two decades (Yeh et al., 2018; Schumacher et al., 2022; Wang et al., 2023; Zhang et al., 2024). Our results (Figs. 1, 4, 7) demonstrate that MELT effectively captures these heatwaves, clearly distinguishing between humid and dry conditions and identifying specific regions experiencing each type. Notably, the 2021 Pacific Northwest (PNW) and the 2016 South Korea heatwaves were identified as being more humid compared to the relatively drier 2019 Western Europe heatwave.

Our findings align closely with studies by Mo et al. (2022) and (Yeh et al., 2018), which identify the humid nature of the PNW and South Korea heatwaves. Specifically, Mo et al. (2022) attribute increased moisture during the PNW heatwave to an atmospheric river, enhancing regional humidity and latent heating. Similarly, Yeh et al. (2018) link enhanced convective activity to elevated moisture content during the South Korea heatwave. Conversely, the Western Europe heatwave was comparatively drier, largely due to dry-air intrusion from the northern Sahara (Sousa et al., 2020). MELT effectively captured these differences, reflected clearly in lower index values for this event.

These results underscore several advantages of the MELT index, particularly its reliance on readily available temperature and relative humidity datasets:



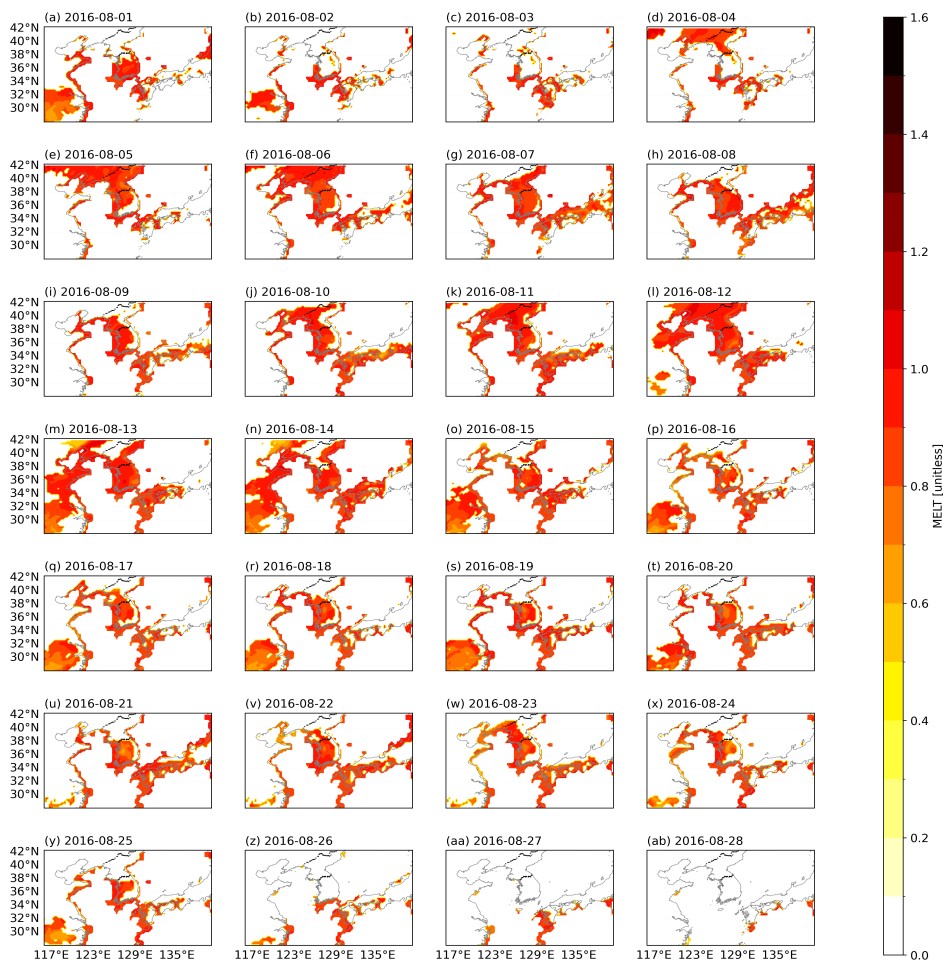

**Figure 7.** **Moisture ELevated Temperature (MELT) Index for the South Korea heatwave in August 2016. Figure shows the MELT index for the heatwave.**



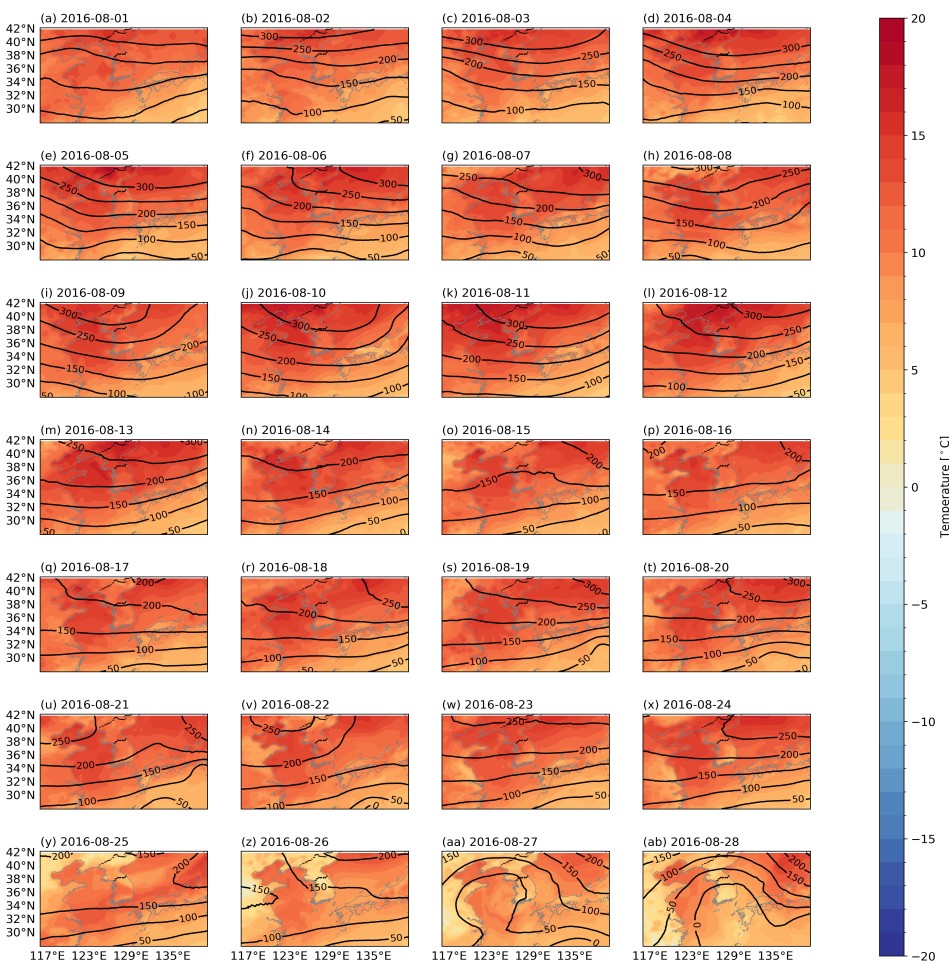

**Figure 8.** Synoptic scale composites of anomalous maximum surface temperature and geopotential height contours (in meters) at 500 hPa for the South Korea heatwave in August 2016.



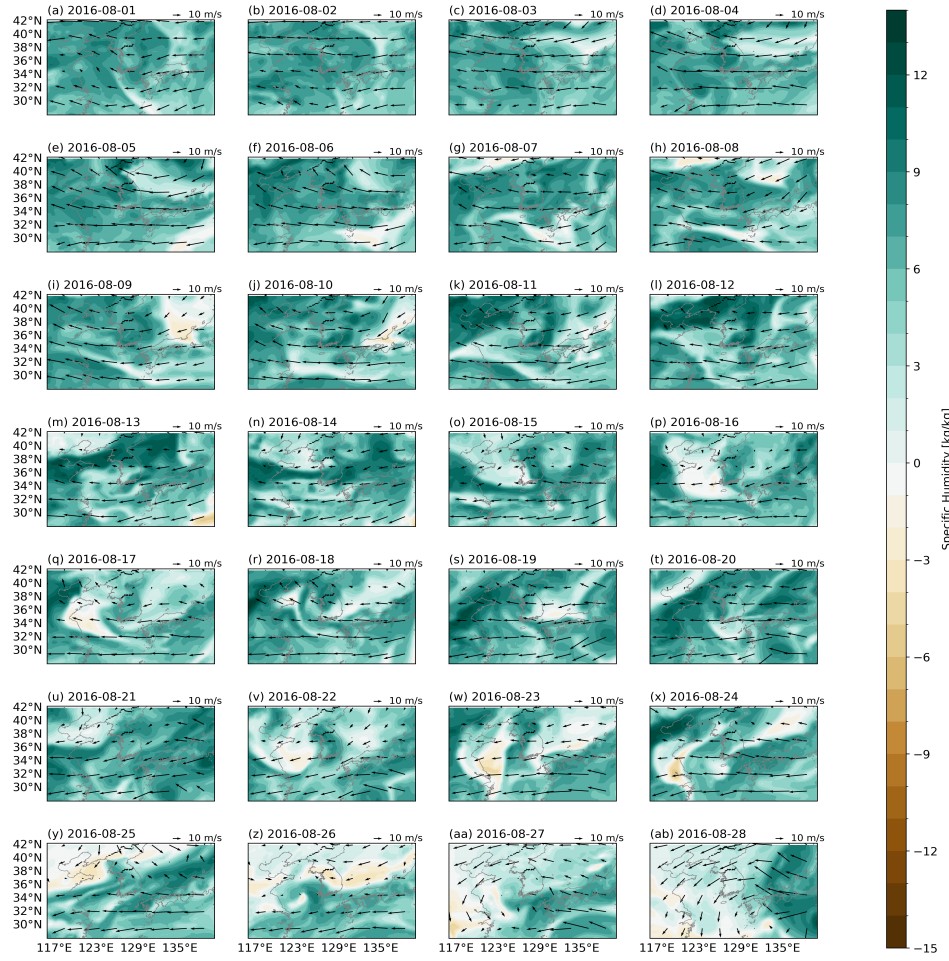

**Figure 9.** Synoptic scale composites for anomalous near-surface (850 hPa) specific humidity and wind circulation (500 hPa $\vec{U}$) for the South Korea heatwave in August 2016.



- It reliably identifies heatwave events and clearly distinguishes humid from dry conditions.
- It provides a global, consistent framework for analyzing heatwaves, overcoming limitations associated with traditional temperature-only indices (Stefanon et al., 2012; Nairn & Fawcett, 2015; Russo et al., 2016; Perkins et al., 2012; Perkins-Kirkpatrick & Lewis, 2020).

Given projections of increased heatwave-related mortality in a warming climate (Åström et al., 2011; Xu et al., 2016; Guo et al., 2017), accurately characterizing heatwaves and understanding their moisture characteristics becomes increasingly critical. Although our study confirms MELT's utility in the current climate, the use of climatological $95^{th}$ percentile thresholds ensures its adaptability to varying climatic contexts.

Future research should explore how heatwave characteristics evolve under climate change, specifically addressing crucial questions such as:

- What is the likelihood of current heatwave events occurring under future climate scenarios?
- Will heatwaves become both warmer and more humid or warmer but drier in future climates?

Addressing these questions will provide essential insights for climate adaptation strategies, mitigation planning, and informed policy-making.

**Open Research Section**

The ECMWF Reanalysis v5 data for this study is publicly available at `https://cds.climate.copernicus.eu/datasets`.

**Acknowledgments**

This work was supported by the Environmental Resilience Institute, funded by Indiana University's Prepared for Environmental Change Grand Challenge initiative and in part by Lilly Endowment, Inc., through its support for the Indiana University Pervasive Technology Institute. Support for K. A. Q. was provided by NSF National Center for Atmospheric Research (NSF NCAR), which is a major facility sponsored by the National Science Foundation (NSF) under Cooperative Agreement No. 1852977.



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
