# Peer review of "Moisture ELevated Temperature (MELT) index: A novel index to capture dry and humid heatwaves"

_EGUsphere, 2025_

## Community Comment (CC3)

**RESPONSE TO CC1**

There are more than 150 heat metrics, including many metrics based on synoptic classifications. The first heatwave early warning system was based on synoptic classifications; many others followed worldwide. Please provide a comparison and contrast of how the proposed metric differs from synoptic classifications in use.

**Response**

We thank the community member for the comment. We agree with the community member that there are many heat metrics, hence, a comparison between metrics might be warranted. We have added some more information to the introduction section to address this concern.

Relative humidity is associated with temperature. Explanations would be helpful as to how this was considered and why absolute humidity was not considered.

**Response**

We thank the community member for the comment. We have added a justification for choosing relative humidity (RH) over absolute humidity (AH) in the revised manuscript. We reiterate this here:

> "We use relative humidity because it provides a direct measure of the moisture content of the atmosphere relative to its capacity to hold moisture, which makes it more intuitive and accessible to the general public. Unlike absolute humidity, RH is widely used in public health communications and is more directly linked to human thermal perception, as it influences the body's ability to cool itself through sweat evaporation at a given temperature."

Citations

Nguyen, J. L., Schwartz, J., & Dockery, D. W. (2014). The relationship between indoor and outdoor temperature, apparent temperature, relative humidity, and absolute humidity. Indoor air, 24(1), 103-112.

Lowen, A. C., Mubareka, S., Steel, J., & Palese, P. (2007). Influenza virus transmission is dependent on relative humidity and temperature. PLoS pathogens, 3(10), e151.

The manuscript needs to make a compelling case that a new metric provides novel insights into heatwave characterization. The manuscript also needs to provide a compelling case that the proposed metric better characterizes physiological heat stress, considering that low and high humidity affect human physiological response to exposure to high ambient temperatures.

**Response**

We thank the community member for the comment. We have revised the manuscript in several places to provide a nuanced compelling case that the metric brings new insight into heatwaves.

It is not accurate to state that epidemiological studies have focused primarily on temperature. Many publications used temperature-humidity indices.

**Response**

We thank the community member for the comment. We have revised the manuscript to describe the idea better.

The metric was tested in three extreme heatwaves, but no criteria were provided for how those heatwaves were selected over the hundreds of other heatwaves that occurred over the past decade.

**Response**

We thank the community member for the comment. We have added the criteria for selecting these specific events. We select these specific events because they their return periods were small (~1 in a 50 year event). We have added this idea to the revised manuscript.

Better characterization of heatwaves does not necessarily translate into improved responses. The 2021 PNW heatdome was accurately forecast, but was so extreme that authorities did not believe it would occur as forecast.

**Response**

We thank the community member for the comment. This is well noted in the revised manuscript.

Many low-resource settings often have limited available temperature and humidity data at the scales needed for decision-making.

**Response**

We thank the community member for the comment. Specifically, we have added this to the discussion section and written that:

> "Although the MELT index offers several advantages, one immediate limitation is its sensitivity to coarse spatial resolutions. At coarser resolutions, the index may fail to adequately capture fine-scale regional features, reducing its usefulness for localized policy-making and decision support."

The cutting-edge research on heatwave early warning and response systems is developing impact-based forecasts. It would be helpful to discuss the value of forecasts using the new metric vs. impact forecasts.

**Response**

We thank the community member for the comment. We add a discussion on the value of this metric relative to impact forecasts, and specifically we add that:

"Additionally, while the MELT index does not directly provide information for impact assessment, it effectively indicates the expected magnitude of a heatwave based solely on atmospheric data, without requiring health or population impact information."

---

## Author Comment (AC1)

**RESPONSE TO RC2**
This is a clearly presented and succinct paper on an interesting topic. However, in my opinion there are some major issues that must be addressed before the paper is considered for publication. I am in full agreement with concerns already raised by the other reviewers, and I provide my own major and minor issues below. The key point (relevant to all my major issues – and raised by the other reviewers) – is that the virtues of the new humid-heat metric are not at all clear. This is very problematic because using relative humidity (RH) over an absolute measure of humidity (like the absolute or specific humidity, or the vapour pressure) is not the obvious choice from an impacts' perspective; it also challenges interpretation of 'dry' and 'moist' heat events. The lack of context regarding how the MELT index is an improvement on other work also does not help convince of the benefits provided by the new index.

**Major issues**

1.  There is a misrepresentation of the extent to which previous work has engaged with this topic. For example, please see Matthews et al. (2022) and Ivanovitch et al. (2024). The former discusses the use of equivalent temperature (and the 'latent' temperature) as a physical quantity to characterise humid heat; the latter presents a novel new metric to communicate the extent to which humidity (and temperature) contribute to a specific level of humid heat (e.g., a given wet-bulb or equivalent temperature, noting that equivalent temperature is (MSE-gz)/Cp). How does the manuscript advance on such work (noting that the above are not empirical metrics/tailored for regional applications)?

    **Response**

    We thank the reviewer for the insightful comment. We acknowledge that the aforementioned studies have made significant advances in the field, particularly through multi-region analyses and the use of non-empirical metrics. To complement and further this progress, our proposed index is designed with the following advantages in mind:

    -   **Ease of calculation** – It is simple to compute using readily available atmospheric variables.
    -   **Accessibility** – It is applicable in data-sparse regions where observational data, such as wet-bulb temperature, may be unavailable. In such cases, reanalysis data can be used effectively.
    -   **Computational efficiency** – The index is computationally inexpensive, making it suitable for large-scale or real-time applications.
    -   **Flexibility** – It can be adapted to different climatological benchmarks, such as the 95th percentile thresholds across varying climate regimes.

    Our goal is not to replace existing methods, but to provide an additional, practical tool for identifying and analyzing heatwaves, particularly in regions with limited resources. We have made modifications in various parts of the manuscript to refine it succinctly and convey these points.

2.      The MELT index might, at least in theory, miss extreme humid events due to the initial dependence on dry-bulb temperature to identify a heatwave. For example, Tmax

might not be that extreme yet, because of very high RH, the wet-bulb temperature could be.

**Response**

We thank the reviewer for the insightful comment. We acknowledge the reviewers' concerns; however, the literature has shown that RH has been used in the health sector and has been identified to correlate well with how people feel. Hence, although this might be a limitation, the index is still able to quantify heatwaves, although it might not be entirely applicable to heat stress

3.      Despite the claims in the paper, I don't think you can compare/characterise 'humid' and 'dry' heatwaves using MELT. Under the authors' definition, a heatwave could be interpreted as dry (e.g., MELT<1 due to relatively low RH) – or at least drier than during a heatwave in another region -- even if the specific humidity was extreme due to very high dry-bulb temperature (and hence higher saturation specific humidity). The same issue is there if we compare heatwaves at a single site. To illustrate, imagine two heat waves (a and b) in the same place with identical specific humidity. If b had higher dry-bulb temperature than a, its MELT would be lower (because saturation specific humidity, and hence RH, would decline). Yet, from a human heat stress perspective, we would expect the cooling potential via sweating to be very similar between heatwaves a and b because that depends on specific/absolute (and not relative) humidity. There are arguments for using RH rather than an absolute measure of humidity; please elaborate if this remains the choice for the MELT index.

**Response**

We thank the reviewer for raising an important and nuanced point about the interpretation of MELT and its relationship with heatwave humidity characteristics.

We agree that MELT, by incorporating relative humidity rather than specific (absolute) humidity, can sometimes lead to counterintuitive characterizations of humid vs. dry heatwaves, particularly in the cases described by the reviewer. As noted, relative humidity decreases with rising temperature when specific humidity remains constant, and therefore, two heatwaves with identical specific humidity but different temperatures may indeed be assigned different MELT values. Our intention in using MELT is to adopt a physically-grounded index that incorporates both temperature and humidity to estimate how humid an atmosphere is at the point when a heatwave is occurring. However, we acknowledge that MELT's dependence on RH can obscure the role of specific humidity in some cases, especially in extremely hot environments where RH naturally declines at a constant specific humidity.

To address this:

- We have revised the manuscript to clarify that MELT does not provide a direct classification of 'dry' versus 'humid' heatwaves in terms of specific humidity, but rather reflects the moisture content of the atmosphere during a heatwave event. Since RH is related to saturation, and hence, higher RH values could be a proxy for classifying how humid it is.
- We have added a brief discussion weighing the merits of RH-based versus specific humidity-based indices.

- Also, our analysis of the specific humidity in the synoptic scale section aligns with the results evinced in using RH in terms of region of high moisture relative to regions of low moisture contents

We appreciate the reviewer's detailed scenario involving two heatwaves at a single location and have included a similar illustration in the revised manuscript to underscore this limitation.

Minor issues

Not all references appear correctly (i.e., only some are hyperlinked). Please check.

**Response**

We thank the reviewer for the comment. We have rectified this in the revised manuscript. In-text references are not hyperlinked, but every other reference is hyperlinked, just as it should be.

*"If an event has a MELT index < 0.5 then, there is a relatively dryer [sic] heatwave with humidity lower than 50% of the climatological relative humidity"...* Not quite, if I understand correctly – it'd be lower than the 47.5th % percentile? (Because the RH is 50 % of the 95th percentile?). Please correct or explain.

**Response**

We thank the reviewer for the comment. The reviewers' understanding is correct. This would mean the RH is 50% of whatever baseline climatology is used. That is, if the 90th percentile is used, then it would be 50% of that (45%). And this is a unique feature of the index which shows that if a different climate regime is used, the MELT index would charcterize heatwaves based on the climates regime and spatial representation of temperature and relative humidity.

Correct 'dryer' to 'drier' (I think it only occurs in one place (above), but please check).

**Response**

We thank the reviewer for the comment. We have made these changes in the manuscript.

References

Matthews, T., Byrne, M., Horton, R., Murphy, C., Pielke Sr, R., Raymond, C., Thorne, P. and Wilby, R.L., 2022. Latent heat must be visible in climate communications. *Wiley Interdisciplinary Reviews: Climate Change*, *13*(4), p.e779.

Ivanovich, C.C., Sobel, A.H., Horton, R.M. and Raymond, C., 2024. Stickiness: A new variable to characterize the temperature and humidity contributions toward humid heat. *Journal of the Atmospheric Sciences*, *81*(5), pp.819-837.

---

## Author Comment (AC2)

**RESPONSE TO RC1**

The MELT index, Moisture Elevated Temperature, is designed to evaluate potential heatwaves and diagnose dry and moist heat events. Categories range from 0 (no heat event), <0.5 (dry enhanced), to ≥1 (moist elevated) scale. The intent is for use to help meteorologists and decision makers to determine the threat to health outcomes. The manuscript then uses 3 recent heatwaves that were recently scrutinized by multiple researchers to showcase the metric. Two of the events, the 2021 Pacific Northwest Heatwave and the 2016 South Korean heatwave were consistent with elevated moist heatwaves. The 3rd event, the 2019 European heatwave, was identified as a dry heatwave. Overall, a straightforward synoptic scale analysis was conducted, and the metric shows promise in meteorological applications.

From the scientific objectives and demonstration, the manuscript is well written and straightforward. I appreciate the authors efforts in readability! From this standpoint, the paper is good, and some minor adjustments which I detail below would be fine.

**Response**

We thank the reviewer for these very constructive comments. We have addressed the reviewer comments in this response and have made modifications to the manuscript to prepare the manuscript for a larger audience with more practical application.

However, there are 3 a fundamental 'elephant-in-the-room' aspects to the paper from an editorial point of view that perhaps should be addressed.

1. The authors discuss physiological responses to moist heat, and that their metric can identify a moist or dry heatwave. But without showing human responses to the moist or dry heatwave examples, it can be hard to determine whether their classification would correctly address the societal outcomes from these heatwaves.

   **Response**

   We thank the reviewer for their thoughtful comment. We agree that our proposed index, MELT, does not directly capture physiological outcomes of heatwaves. Rather, the primary aim of our index is to serve as a physically-based tool that can be used **in conjunction with health outcome data and other heat stress metrics** to better define and characterize heatwaves.

   We acknowledge that MELT is not a direct measure of human heat stress response. However, one of its key strengths lies in its use of readily available meteorological data (air temperature and relative humidity) which allows for broad application, including in **data-sparse regions** where more complex indices may not be feasible.

   To address the reviewer's concern, we have clarified throughout the manuscript that **we do not claim MELT to be the best or universal index**, but rather a complementary method for identifying multi-regional heatwave regimes within a consistent climatological framework. We have added language in several key sections (Introduction, Discussion, and Conclusions) to highlight this nuance and emphasize that MELT should be interpreted as one component within a broader toolkit of heatwave and heat stress metrics.

We believe that MELT offers a useful approach for regional climate studies, particularly in contexts where observational constraints limit the use of more sophisticated heat stress measures, and appreciate the opportunity to clarify this point.

2.      There are 100s of heat stress metrics. The authors mention that metrics are not quite universal or are region-specific. But this gets into the nuances of heat stress: there are 100s of heat stress metrics because humans respond to heat in a myriad of ways. A dry heatwave can be just as deadly as a wet heatwave. Exposure, duration, activity, health status, age, etc., all interplay into heat stress. It is very difficult to generalize heat stress on humans to an individual index. Fundamentally, this is because the thermo-physiological system is complex, and there are many 'paths' that lead to negative outcomes, many of which are non-linear. So, I am skeptical that this index would be universally applied. What is interesting is the interplay with meteorology, which is clearly demonstrated from the authors' synoptic analysis.

**Response**

We thank the reviewer for this comment. We agree that multiple heat stress metrics exist, reflecting the fact that humans respond to heat in diverse ways, and it is therefore difficult to generalize a single metric for all contexts. While the current form of our manuscript may have implied otherwise, that was not our intention.

The goal of our proposed method is not to replace existing metrics, but to provide a novel, simple, efficient, and computationally accessible approach to characterizing heatwaves, particularly by using a specific climate regime as a benchmark. We have clarified this point in the revised manuscript to better motivate the rationale behind the development of this index. Specifically, we have added the following to the text:

*"Furthermore, in regions where observational data is sparse or unavailable, it can be challenging to quantify certain heatwave or heat stress metrics. For example, the use of wet-bulb temperature, which is a commonly used indicator for heat stress, involves complex computations and approximations (e.g., Stull 2011) that may not be readily accessible, particularly in developing countries. Our method is not intended to replace existing approaches, but rather to complement them. It offers a more accessible and practical means of identifying heatwaves by benchmarking against a reference climate regime, especially in data-scarce regions or where computational resources are limited."*

3.      Lastly, there is the aspect of applying MELT to future climate change. Buzan and Huber, 2020 show that relative humidity scales negatively with global mean temperature change. This does not mean that future heatwaves would switch from moist to dry in the future. There are a lot of issues with relative humidity as a basis for a metric, and one of those is that absolute humidity increases non-linearly with temperature. Even if extreme relative humidity goes down in the future, the danger from the heatwave is enhanced due to the unusually elevated moisture.

**Response**

We thank the reviewer for highlighting that some studies (e.g., Buzan & Huber 2020) report a negative trend in relative humidity (RH) with global warming, which

could bias an RH-based metric toward "dry" classifications. However, recent work demonstrates that conditional RH, i.e., RH given a particular high temperature, often increases in a warming climate. For example, Matthews et al. (2018) demonstrated that both temperature and RH exhibit an upward trend in a warming climate. Similarly, Yuan et al. (2020) showed that *"in a warming climate, a future day will tend to have higher RH than a day of the same temperature under the historic climate."*

Consequently, even at the same temperature, increased RH in the future will amplify heat stress. Ignoring this conditional increase in RH, as done in some recent assessments of heat impacts, may lead to underestimating the severity of future heatwaves. Therefore, the observed negative scaling between temperature and RH in certain studies may reflect regional or temporal variability, rather than a consistent global response to rising temperatures. Moreover, the Clausius-Clapeyron relation suggests that with increasing temperature, the atmosphere's capacity to hold moisture increases, which generally supports a rise in humidity.

**Citations**

Yuan, J., Stein, M. L., & Kopp, R. E. (2020). The evolving distribution of relative humidity conditional upon daily maximum temperature in a warming climate. Journal of Geophysical Research: Atmospheres, 125(19), e2019JD032100.

Matthews, Tom. "Humid heat and climate change." Progress in Physical Geography: Earth and Environment 42.3 (2018): 391-405.

To me, what I discuss in the 3 paragraphs above is an editorial decision on the status of the manuscript, because, as stated before, the paper is well written and is self-contained with clear scientific objectives, construction, and application. Below are the minor comments that should be addressed.

Best regards,

-Jonathan R. Buzan

**Response**

We thank the reviewer for these insightful and constructive comments. We have revised the manuscript with the above suggestions in mind.

Line 10: may want to remove the mention of accurately assessing physiological heat stress. The paper does not compare with health data. Emphasize the meteorological applications instead.

**Response**

We thank the reviewer for the comment. In the context of this statement, we are stating the importance of moisture and temperature for physiological heat stress and not

necessarily the MELT index. We have revised this statement to clearly articulate the intent of the sentence.

Line 42—50: Buzan et al., 2015 shows that batteries of heat stress metrics cover a larger swath of societal outcomes. Furthermore, the manuscript also comes up with methods that address the dry vs wet heat through this battery of metrics. The utility of using multiple metrics also allows for broader applications, such as the interplay of infrastructure, climate change, and heat stress (Parkes et al., 2022). Additionally, Ivanovich et al., 2024 also created a new index called 'stickiness' that also goes into splitting heatwaves into dry and wet classifications. These manuscripts should be mentioned and discussed.

**Response**

We thank the reviewer for the comment. We have added further discussion on these papers as described by the reviewer in the revised manuscript.

Line 59-66: WBGT and UTCI go one step further than temperature-humidity covariance, they also include radiation… as long as they are calculated correctly (Cvijanovic incorrectly calculates WBGT, even with the assumptions about "radiation free" environment). Buzan, 2024 highlights the temperature-humidity-radiation relationship.

**Response**

We thank the reviewer for the comment.

Line 86: The reference period includes the climate events. Does this change when choosing a different reference period? Or climate change?

**Response**

We thank the reviewer for the comment. This is one interesting aspect of the MELT index. When using this index, a preferred climate regime can be selected for inference and heatwaves calculated using that period. Hence, we would expect that a different reference period would yield heatwaves that are characteristic of that reference regime. This makes the MELT index applicable across different climate regimes.

Line 100-106: The temporal resolution of RH is not stated, and daily maximum is not stated for temperature. I recommend making each step explicit on what data is used. I found it confusing. Buzan and Huber, 2020 and Buzan, 2024 demonstrate that changes in precision can change the outcomes.

**Response**

We thank the reviewer for the comment. In the manuscript, we give detail to the dataset and temporal resolution used in this work in the "Synoptic, dynamics and thermodynamics drivers associated with MELTS" section. We have further added the temperature and relative humidity resolutions to the methods section as well.

Figure color bars: use less colors, especially with the MELT figures. The patterns should become easier to see. For example, Figures 4 and 6. The color bars here become important. There are a lot of sign changes for specific humidity, but it looks like the elevated specific humidity corresponds with the wet heat in the MELT. I was a little confused by this. It will likely become clearer with less color steps.

**Response**

We thank the reviewer for the comment. We have addressed this in the revised manuscript.